# Loss of Muscle Mass and Vulnerability in Institutionalized Older Adults

**DOI:** 10.3390/ijerph20010426

**Published:** 2022-12-27

**Authors:** Mayara Priscilla Dantas Araújo, Thaiza Teixeira Xavier Nobre, Clara Wilma Fernandes Rosendo, Flávio Anselmo Silva de Lima, Vilani Medeiros de Araújo Nunes, Gilson de Vasconcelos Torres

**Affiliations:** 1Graduate Program in Health Sciences, Federal University of Rio Grande do Norte (UFRN), Natal 59078-970, RN, Brazil; 2College of Health Sciences of Trairi, Federal University of Rio Grande do Norte (UFRN), Santa Cruz 59200-000, RN, Brazil; 3Graduating in Medicine, Federal University of Rio Grande do Norte (UFRN), Natal 59078-970, RN, Brazil; 4Graduating in Public Health, Federal University of Rio Grande do Norte (UFRN), Natal 59078-970, RN, Brazil; 5Department of Public Health, Federal University of Rio Grande do Norte (UFRN), Natal 59078-970, RN, Brazil; 6Nursing Department, Federal University of Rio Grande do Norte (UFRN), Natal 59078-970, RN, Brazil

**Keywords:** aged, nutrition assessment, health vulnerability

## Abstract

This study aimed to evaluate the association between muscle mass and vulnerability in institutionalized older adults. A cross-sectional study was carried out in eight philanthropic Long-Term Care Facilities (LTCF) located in the metropolitan area of Natal, Rio Grande do Norte, Brazil. The participants were individuals aged 60 years or older who were present in the institutions at the time of data collection. To assess muscle mass, the calf circumference was categorized into loss of muscle mass (<31 cm) and preserved muscle mass (≥31 cm). The vulnerability was assessed by The Vulnerable Elders Survey (VES-13). Of the 250 older adults evaluated, 46.1% presented loss of muscle mass, which was associated with the presence of physical limitation, vulnerability, and age group (*p* < 0.05). The presence of vulnerability was the main factor contributing to loss of muscle mass (R^2^ = 8.8%; B = 0.781; 95% CI 0.690–0.884; *p* < 0.001). Loss of muscle mass is associated with disability in institutionalized older adults.

## 1. Introduction

Aging leads to changes in the quantity and quality of muscle mass, which directly impacts the functional capacity of individuals [1]. These changes are characteristic of sarcopenia, a skeletal muscle disorder that is associated with falls, fractures, physical disability, and mortality [2]. Sarcopenia is prevalent in older adults, especially in institutionalized people [3,4]. The Older Adult Health Handbook [5] recommends the measurement of the calf circumference as a quick and simple way to monitor whether there is a loss of muscle mass and, thus, as a good marker for evaluating muscle mass in this population [6,7]. 

With the decrease in muscle mass that occurs with advancing age [8], older adults become more susceptible to adverse events to their health, which can be enhanced by the presence of vulnerability. Although this condition is characterized as a public health problem because it results in unfavorable situations to the health of older adults, such as disabilities and physical limitations [9], it is still little explored in the context of Long-Term Care Facilities (LTCF). Vulnerability, an important factor in the multidimensional evaluation of older adults included in the Vulnerable Elders Survey-13 (VES-13), identifies older adults at high risk of functional decline and mortality in two years [10]. This instrument evaluates the functional performance of older adults, an important factor related to the level of sarcopenia that helps predicting adverse results [2].

A study found an inversely proportional association between calf circumference and frailty in older adults [11]. However, it is not yet known whether the same is true for vulnerability, especially in institutionalized older adults. Knowing whether this association is present or not will contribute to the proposition of actions to minimize muscle mass loss and functional decline. The timely identification of vulnerability can delay the worsening of the functional decline and contribute to greater independence, autonomy, and quality of life in older adults [12].

Although the loss of muscle mass and vulnerability are associated and frequent conditions in the older population, the relationship between them is still little explored in the literature. This study hypothesizes that the loss of muscle mass is associated with the presence of vulnerability in institutionalized older adults. Therefore, this study aimed to evaluate the association between muscle mass and vulnerability in institutionalized older adults.

## 2. Materials and Methods

This is a cross-sectional study with a quantitative approach through the analysis of primary data of 267 older adults living in eight philanthropic non-profit LTCF, six located in the municipality of Natal, one in Parnamirim, and one in Macaíba, in the state of Rio Grande do Norte, Brazil. 

Data were collected by a multidisciplinary research team composed of nurses, physiotherapists, and nutritionists between February and December 2018. The information on the sociodemographic and health characterization and anthropometric data of the older adults was taken from their medical records and the Older Adult Health Handbook, 2017 version [5], recommended by the Ministry of Health in Brazil as a screening tool for the multidimensional evaluation of older adults.

This study adopted convenience sampling (non-probabilistic). Institutionalized older adults aged 60 years or older who were present in the institutions at the time of data collection were included. The information of bedridden older adults and those with dementia included in this study was obtained from their medical records and from the Older Adult Health Handbook, previously completed by the professionals of the LTCF. 

The sociodemographic characteristics collected and analyzed were: sex (male; female), age group in years (60 to 74; 75 to 84; ≥85), race/color (white; non-white), marital status (single; widowed; divorced/separated), literacy (yes; no), schooling (none; 1 to +8 years), religion (yes; no), and which religion (catholic; evangelical; spiritist). The lifestyle habits analyzed were smoking (yes; no) and alcohol consumption (yes; no).

The body mass index (BMI) and calf circumference (CC) were used to assess nutritional status. The body mass index was calculated by dividing the weight in kilograms (kg) by the height and reported in m^2^, categorized as low weight (<22 kg/m^2^), adequate weight (≥22 kg/m^2^ and <27 kg/m^2^), and overweight (≥27 kg/m^2^) [5]. The Older Adult Health Handbook [5] cutoff point was used for categorization of CC, in which CC <31 cm is considered an indicative of loss of muscle mass. Thus, CC values were categorized as follows: loss of muscle mass (<31 cm) and preserved muscle mass (≥31 cm). This categorization did not differ for sex. In the case of older adults with restricted mobility, such as the bedridden and wheelchair users, the anthropometric variables collected from the medical records were previously estimated by the nutrition professionals of the LTCF.

Vulnerability was assessed by the Vulnerable Elders Survey (VES-13) in the adapted [13] and validated [14] version for use in the Brazilian older population. This instrument consists of 13 items divided into four domains: age (one point for 75–84 years; three points for ≥85 years); self-perceived health (one point for fair or poor); physical limitation, which assesses the difficulty to stoop, crouch or kneel, lift or carry objects weighing 5 kg, reach or extend arms above shoulder level, write or handle and grasp small objects, walk 400 m and do heavy housework (one point per activity with “too much difficulty” or “unable to do”, with a maximum score of two points), categorized in absence and presence of difficulty; and disability, which assesses whether the person needs help to shop for personal items, manage the own money, walk across the room, do light housework, and bathe or shower alone (four points per answer “yes”, with the maximum score of four points). The score obtained in each domain is summed and the final value can reach a maximum score of 10 points. When the value ≥3 points, the older adult is considered vulnerable. 

The collected data were tabulated in the Excel 2010^®^ (Microsoft Office) and analyzed in the Statistical Package for Social Sciences (SPSS) version 21.0 for Windows. Data normality was evaluated by the Kolmogorov–Smirnov test and a non-normal distribution was detected. Descriptive statistical analysis was performed to characterize the sample and vulnerability from the VES-13 domains using absolute and relative frequencies, means, standard deviation, minimum and maximum values, and percentiles (25/50/75), bivariate analysis to evaluate the association of independent variables with CC, using Pearson’s Chi-Square, Fisher’s Exact, Mann–Whitney and Logistic Binary Regression tests, considering the 95% confidence interval. The variables that presented a significance level of 5% (*p* < 0.05) in the bivariate analysis were considered significant because they were associated with CC.

This study met the ethical precepts of research with human beings and was submitted for consideration to the Research Ethics Committee of the Onofre Lopes Hospital (CEP/HUOL) and approved under Opinion number 2,366,555 and CAAE: 78891717.7.0000.5292. To carry out the study, the participants or guardians were requested to give consent and sign the informed consent form (ICF).

## 3. Results

The study population consisted of 267 institutionalized older adults, of whom 250 (93.6%) had data regarding CC, thus composing the final sample of this study. Of the 250 participants, 170 (68.0%) presented complete data regarding the variables studied.

The sociodemographic characteristics, lifestyle habits, and nutritional status of the older adults evaluated are presented in Table 1. Among them, 46.1% presented loss of muscle mass. It was observed that the loss of muscle mass was more frequent in women, aged 85 years or older, single, non-white, with higher schooling, non-smokers and/or not consumers of alcohol, and who had low weight. Statistical associations were found between loss of muscle mass and age equal to or greater than 85 years (*p* = 0.006), non-smoking (*p* = 0.005), and low weight (*p* < 0.001).

Table 2 shows the sub-items of the physical limitation and disabilities domains regarding the presence or absence of difficulty and the performance of activities of daily living. Loss of muscle mass was more frequent in older adults with difficulty doing heavy housework and stooping, crouching, or kneeling (84.4%). The presence of difficulty in performing activities involving the lower and upper limbs was associated with loss of muscle mass in institutionalized older adults (*p* < 0.05). It was observed the predominance of loss of muscle mass in the presence of disabilities, which were significantly associated (*p* < 0.05). These findings demonstrate the association between decreased autonomy and independence of older adults with the worst nutritional status.

The older adults with loss of muscle mass showed a higher total score in the VES-13 domains, except in the self-perceived health. In the disability domain, although the *p*-value was >0.05, there was a trend of higher scores in the presence of loss of muscle mass (Table 3). 

The binary logistic regression test was performed to determine the extent to which the loss or maintenance of muscle mass explains the score obtained in the VES-13. The loss of muscle mass explains in 8.8% the cases of vulnerability (Table 3).

## 4. Discussion

In this cross-sectional study, the association between muscle mass and vulnerability in institutionalized older adults was evaluated, and a predominance of muscle mass loss was significantly associated in older adults with physical limitations and disabilities. Older adults with loss of muscle mass obtained higher scores in the VES-13, except in the self-perceived health domain, and vulnerability was the factor that most contributed to the loss of muscle mass.

Loss of muscle mass was predominant in women, in people aged between 75 and 84 years, and with low weight, which are factors independently associated with sarcopenia in institutionalized older adults [3], but commonly observed in LTCF [15,16,17]. Studies suggest that the hormonal changes experienced by women during the climacteric period lead to a reduction in muscle and bone mass [18,19]. Added to this, with advancing age, there is a nutritional decline, characterized mainly by the loss of muscle mass [20].

Malnutrition and loss of muscle mass were concomitant in most of the older adults evaluated, as it was also observed in institutionalized Spanish older adults [21]. Malnutrition contributes to the loss of muscle mass [22], a condition that leads to a reduction in muscle strength and, altogether, they increase the risk of mortality in older adults [23]. Nutritional decline also directly affects the functionality of these people, and, thus, adequate protein consumption is suggested as a strategy to reduce loss of muscle mass and strength [8]. 

Physical limitations occur due to dependence resulting from functional decline and are associated with the institutionalization of older adults [24], which could explain the predominance of older people with difficulties in performing daily life activities. Studies have identified an association between the presence of physical limitations and a worse nutritional status in institutionalized older adults [25], in addition to a higher risk of disability in sarcopenic individuals [26]. The results of the present study show that there is an association between the loss of muscle mass and the presence of physical limitation, although a trend towards the association of this loss with the presence of disability was observed and could reach statistical significance if we had a larger sample. These findings suggest that maintaining an adequate nutritional status can positively impact the functional capacity and autonomy of institutionalized older adults, and multidimensional care strategies involving the nutritional and functional aspects of these people can be adopted.

The presence of difficulty in performing all the activities that make up the physical limitation domain was associated with the loss of muscle mass. This finding demonstrates the negative impact of muscle mass loss on the independence of older adults, and that muscle health is fundamental to maintaining their functionality. The institutionalized older adults hardly reach nutritional recommendations through food consumption [15], and this is harmful to their health. In a study with institutionalized Canadian older adults, an association between diet quality with low weight and lower calf circumference was found [27]. Inadequate nutritional intake can lead to the development of frailty syndrome [28] and sarcopenia [29]. 

Loss of muscle mass is associated with the presence of vulnerability in institutionalized older adults. In a study with older Polish adults, the participants with sarcopenia obtained three or more points in the VES-13 [30], indicating the presence of vulnerability. This instrument is a good predictor of mortality, disability, and institutionalization [31]. According to the average scores obtained by the institutionalized older adults in the present study, it is evident that they experience a greater worsening of vulnerability and are, therefore, at higher risk of adverse health events. This shows the need to intervene in the modifiable factors that lead to this condition, such as maintaining the functionality of these people, aiming at their independence and autonomy, and adopt strategies for maintaining their muscle mass. Thus, we suggest the development of studies that will identify which strategies can be adopted to minimize the worsening of vulnerability and its consequences on the health of institutionalized older adults.

There is a potential limitation in this study that could be addressed in future research. The Older Adult Health Handbook [5] used for data collection does not differ in the CC cutoff point for men and women. Due to the difference in body composition between men and women, we suggest the improvement of this tool to differ the cutoff points by sex and give a more reliable nutritional status assessment for the older population.

## 5. Conclusions

This study investigated the association between muscle mass and vulnerability in institutionalized older adults. Statistically significant differences were found between loss of muscle mass and age group, presence of physical limitation, and vulnerability. This study demonstrated that vulnerability directly contributes to the loss of muscle mass and that it is necessary to intervene in the modifiable factors that lead to this condition through actions aimed at minimizing the effects of aspects of vulnerability in the institutional environment.

We consider the use of VES-13 as an important tool that provides professionals working in LTCF with opportunities to carry out care plans for the residents aiming to intervene in a timely manner and prevent, delay, treat, and sometimes even reverse the loss of muscle mass and its consequences to the health of older adults.

## Figures and Tables

**Table 1 ijerph-20-00426-t001:** Sociodemographic characterization, lifestyle habits, and nutritional status and their association with the muscle mass of older adults living in LTCF. Natal, RN, Brazil, 2021.

Variables		Loss of MM n (%)	Preserved MM n (%)	Total ^1^ n (%)	*p*-Value ^2^
Demographic					
Sex	Female	91 (36.4)	80 (32.0)	171 (68.4)	0.062
	Male	32 (12.8)	47 (18.8)	79 (31.6)	
Age group in years	60 to 74	25 (10.0)	43 (17.2)	68 (27.2)	0.006
	75 to 84	45 (18.0)	52 (20.8)	97 (38.8)	
	≥85	53 (21.2)	32 (12.8)	85 (34.0)	
Race	Non-white	59 (25.0)	63 (26.7)	122 (51.7)	0.801
	White	57 (24.2)	57 (24.2)	114 (48.3)	
Marital status	Single	64 (27.0)	52 (21.9)	116 (48.9)	0.169
	Widowed	31 (13.1)	40 (16.9)	71 (30.0)	
	Divorced/separated	21 (8.9)	29 (12.2)	50 (21.1)	
Literacy	Yes	62 (26.6)	78 (33.5)	140 (60.1)	0.058
	No	53 (22.7)	40 (17.2)	93 (39.9)	
Schooling	No	48 (21.7)	37 (16.7)	85 (38.5)	0.058
	1 to +8 years	59 (26.7)	77 (34.8)	136 (61.5)	
Religion	Yes	102 (45.7)	101 (45.3)	203 (91.0)	0.193
	No	7 (3.1)	13 (5.8)	20 (9.0)	
Type of religion	Catholic	78 (38.8)	74 (36.8)	152 (75.6)	0.864
	Evangelical	22 (10.9)	25 (12.4)	47 (23.4)	
	Spiritist	1 (0.5)	1 (0.5)	2 (1.0)	
Lifestyle habits					
Smoking	Yes	9 (3.7)	25 (10.2)	34 (13.9)	0.005
	No	110 (44.9)	101 (41.2)	211 (86.1)	
Alcohol consumption	Yes	4 (1.6)	11 (4.5)	15 (6.1)	0.080 ^3^
	No	115 (46.9)	115 (46.9)	230 (93.9)	
Nutritional status					
BMI	Low weight	72 (30.6)	16 (6.8)	88 (37.4)	<0.001
	Adequate	32 (13.6)	45 (19.1)	77 (32.8)	
	Overweight	9 (3.8)	61 (26.0)	70 (29.8)	

MM: Muscle mass; BMI: Body Mass Index. ^1^ Total number of variable searches due to lack of information. ^2^ Pearson Chi-Square Test; ^3^ Fisher’s Exact Test.

**Table 2 ijerph-20-00426-t002:** Association of physical limitation and disability with calf circumference in older adults living in LTCF. Natal, RN, Brazil. 2021.

Physical Limitations and Disabilities		Loss of MM n (%)	Preserved MM n (%)	Total ^1^ n (%)	*p*-Value ^2^
Physical limitation					
Performing heavy housework	Difficulty	110 (45.1)	96 (39.3)	206 (84.4)	0.002
	No difficulty	10 (4.1)	28 (11.5)	38 (15.6)	
Stooping, crouching, or kneeling	Difficulty	104 (42.1)	88 (35.6)	192 (77.7)	0.005
	No difficulty	18 (7.3)	37 (15.0)	55 (22.3)	
Lifting or carrying objects as heavy as 5 kg	Difficulty	95 (38.8)	77 (31.4)	172 (70.2)	0.001
	No difficulty	24 (9.8)	49 (20.0)	73 (29.8)	
Walking 400 m	Difficulty	93 (37.5)	76 (30.6)	169 (68.1)	0.004
	No difficulty	28 (11.3)	51 (20.6)	79 (31.9)	
Reaching or extending arms above shoulder level	Difficulty	66 (26.9)	40 (16.3)	106 (43.3)	<0.001
	No difficulty	53 (21.6)	86 (35.1)	139 (56.7)	
Writing or handling and grasping small objects	Difficulty	64 (26.1)	33 (13.5)	97 (39.6)	<0.001
	No difficulty	55 (22.4)	93 (38.0)	148 (60.4)	
Disabilities					
Shopping for personal items	Yes	107 (44.2)	96 (39.7)	203 (83.9)	0.012
	No	12 (5.0)	27 (11.2)	39 (16.1)	
Managing money	Yes	106 (43.8)	93 (38.4)	199 (82.2)	0.006
	No	13 (5.4)	30 (12.4)	43 (17.8)	
Doing light housework	Yes	95 (39.6)	79 (32.9)	174 (72.5)	0.003
	No	22 (9.2)	44 (18.3)	66 (27.5)	
Bathing or showering	Yes	90 (37.2)	47 (19.4)	137 (56.6)	<0.001
	No	28 (11.6)	77 (31.8)	105 (43.4)	
Walking across the room	Yes	74 (30.6)	41 (16.9)	115 (47.5)	<0.001
	No	44 (18.2)	83 (34.3)	127 (52.5)	

MM: Muscle mass. ^1^ Total number of searched variables due to lack of information. ^2^ Pearson Chi-Square Test.

**Table 3 ijerph-20-00426-t003:** Descriptive analysis and binary logistic regression to evaluate the association between muscle mass and VES-13 scores in older adults living in LTCF. Natal, RN, Brazil. 2021.

VES-13	Loss of MM		Preserved MM		R ^2^ (Exp(B); 95% CI) ^1^ *p*-Value
	Mean (SD)	Percentiles (25/50/75)	Mean (SD)	Percentiles (25/50/75)	
Vulnerability	7.4 (1.98)	6/7/9	6.3 (2.36)	5/7/8	8.8 (0.781; 0.690–0.884) <0.001
*p*-value ^2^	0.001				
Physical limitation	1.7 (0.65)	2/2/2	1.4 (0.84)	1/2/2	7.0 (0.533; 0.376–0.757) <0.001
*p*-value ^2^	0.001				
Age group	1.7 (1.23)	1/1/3	1.2 (1.15)	0/1/3	5.5 (0.709; 0.574–0.876) 0.001
*p*-value ^2^	0.006				
Disability	3.7 (1.10)	4/4/4	3.3 (1.50)	4/4/4	2.8 (0.796; 0.651–0.973) 0.026
*p*-value ^2^	0.052				
Self-perceived health	1.0 (0.00)	1/1/1	1.0 (0.00)	1/1/1	-
*p*-value ^2^	-				

MM: Muscle mass; CI: Confidence Interval; SD: Standard deviation. ^1^ Binary Logistic regression test; ^2^ Mann–Whitney U Test.

## Data Availability

The data presented in this study are available on request from the corresponding authors. The data are not publicly available due to their containing information that could compromise the privacy of research participants.

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
