# Peer review of "Loss of Muscle Mass and Vulnerability in Institutionalized Older Adults"

_ijerph, 2022, doi:10.3390/ijerph20010426_

Round 1

Reviewer 1 Report

We appreciate the opportunity to review this work about the association between the loss of muscle mass and vulnerability. Studies evaluating the functional abilities of older adults are fundamental to improving the well-being of this population and ensuring that older adults live more years and these years will be lived with a good quality of life.

  1. The authors used the Vulnerable Elders Survey (VES-13), is this tool validated for Brazilian institutionalized older adults? Please can insert the reference.

Please revise the text. 

2.      There are spell mistakes  

3.      The text lack uniformization. The authors write older people and elderly, please replace elderly with “older”

4   Is the sample representative?

5.      Can you explain the inclusion criteria? Have you included people with dementia?

6.      In the last decades, the World Health Organization has developed a set of tools to improve the well-being of older adults through the improvement of intrinsic capacities, including motor capacity. Why the authors did not use the WHO guidelines to evaluate the loss of MM?

7.      Vulnerability is associated with other intrinsic capacities and functional abilities such as psychological capacity, vitality, visual capacity, and hearing capacity, number of diseases….

Have you associated the number of medicines used by older adults or the number and type of diseases of older adults with the loss of muscle mass?

8.      The authors concluded:  Loss of MM is associated with disability in institutionalized older people.

In my opinion, the authors should explore if the increased vulnerability caused by several factors ignored in this study (such presence or absence of disease, and polypharmacy) did not influence the results reported.

Reviewer 2 Report

Line 38 replace  "health of the r adults" with "health of older adults". 

line 58 replace "were" with "was" 

Line 62 replace "were" with "was"

Line 85 restriction sums- replace with some restrictions.

In table 1: race should be in the 3rd row and not the word as depicted. replace widower with widowed because they can be male or female. Lifestyle habits instead of life habits as a variable. 

lines 196-198- "Some nutrients have shown beneficial effects on the preservation of muscle strength and delay of functional decline and loss of muscle strength associated with aging, such as proteins, vitamin D, selenium, and magnesium" I recommend to remove this, it is not relevant to this study.
